# Swedish nationwide time series analysis of influenza and suicide deaths from 1910 to 1978

Christian Rück ![ORCID] , David Mataix-Cols, Kinda Malki, Mats Adler, Oskar Flygare ![ORCID] , Bo Runeson, Anna Sidorchuk

Centre for Psychiatry Research, Department of Clinical Neuroscience, Karolinska Institutet & Stockholm Health Care Services, Region Stockholm, Stockholm, Sweden

**Correspondence to**
Dr Christian Rück;
christian.ruck@ki.se

## ABSTRACT

**Objectives** There is concern that the COVID-19 pandemic will be associated with an increase in suicides, but evidence supporting a link between pandemics and suicide is limited. Using data from the three influenza pandemics of the 20th century, we aimed to investigate whether an association exists between influenza deaths and suicide deaths.

**Design** Time series analysis.

**Setting** Sweden.

**Participants** Deaths from influenza and suicides extracted from the Statistical Yearbook of Sweden for 1910–1978, covering three pandemics (the Spanish influenza, the Asian influenza and the Hong Kong influenza).

**Main outcome measures** Annual suicide rates in Sweden among the whole population, men and women. Non-linear autoregressive distributed lag models was implemented to explore if there is a short-term and/or long-term relationship of increases and decreases in influenza death rates with suicide rates during 1910–1978.

**Results** Between 1910 and 1978, there was no evidence of either short-term or long-term significant associations between influenza death rates and changes in suicides (β coefficients of 0.00002, p=0.931 and β=0.00103, p=0.764 for short-term relationship of increases and decreases in influenza death rates, respectively, with suicide rates, and β=−0.0002, p=0.998 and β=0.00211, p=0.962 for long-term relationship of increases and decreases in influenza death rates, respectively, with suicide rates). The same pattern emerged in separate analyses for men and women.

**Conclusions** We found no evidence of short-term or long-term association between influenza death rates and suicide death rates across three 20th century pandemics.

## Strengths and limitations of this study

► To our knowledge, this was the first study to investigate the association between influenza deaths and suicide across several pandemics.
► The large amount of nationwide data on influenza and suicide death rates covering 1910–1978 is a strength of the study.
► To guard against effects of changes in the recording of causes of death, we created a series of dummy variables for each corresponding period, but no significant effects were found.
► No historical data with higher temporal resolution than yearly data could affect the results if the time sequence of association between changes in influenza death and suicide differ from the chosen time interval.

## INTRODUCTION

Various international surveys have documented a negative impact of the COVID-19 pandemic on the population's mental health, with increased levels of psychological stress, psychiatric symptoms, insomnia and alcohol consumption.[1–4] Whether these acute impacts will persist long term is currently unknown. The Royal College of Psychiatrists in the UK and the WHO as well as the International Academy of Suicide Research have raised concerns of a possible increase in suicide rates.[5–7] Such concerns originate from a combination of known risk factors for suicide, including the impacts of social distancing and disconnectedness, an economic downturn, the decreased access to mental health services and increased access to lethal means exemplified by an increase in gun purchases in the USA.[8] For example, a study of the economic recession in USA 2007–2009 found that for every percentage point increase in the unemployment rate, there was about a 1.6% increase in the suicide rate.[9 10] These findings have been questioned. A study using an interrupted time-series analysis taking, for example, seasonality and long-term trends into account, found little evidence that the recession resulted in net excess suicides across all age and gender groups.[11] However, there was some evidence of excess suicides among men aged 65 years and above and young women. Another study failed to find an increase in suicide rates in Sweden during the two most recent economic recessions.[12]

While the concern is widespread, there is currently little evidence to support a clear association between the ongoing COVID-19 outbreak and an increased risk of suicide. Data from the US Centers for Disease Control and Prevention show that suicide decreased in 2020, compared with the preceding year.[13] Data from April to October 2020 from the UK did not show an increase of suicides.[14] The first months of the COVID-19 pandemic in 21 countries was studied by Pirkis and colleagues[15] and overall there was little support for an increase in suicides. Other reports highlight that the outcomes differ and that certain minorities may be at higher risk.[16]

Historic US mortality data from the largest pandemic in the 20th century, the Spanish influenza, showed that the Spanish influenza was associated with an increase in suicides but those effects may have been mitigated by a decline in alcohol consumption.[17] Suicide during the same pandemic in Taiwan, where the Japanese Colonial Government implemented physical distancing, school closings and prohibited religious activities, was studied in a paper by Chang and colleagues.[18] They reported a small and short-lived increase in suicides in the second wave of the pandemic. A study of the impact of social distancing measures on suicide in 1918 in the USA showed that increasing social distancing was associated with increased suicide rates independent of the influenza mortality rate.[19] In Sweden, there were measures in place during the Spanish influenza to minimise the spread of the disease, such as school closings, and public gatherings, cinemas and religious meetings were temporarily stopped in some places.[20]

The outbreak of the severe acute respiratory syndrome (SARS) epidemic in 2003 in Hong Kong was according to two studies associated with an increase of suicide in the population above the age of 65 years.[21 22] In the first of these two reports, the association was only statistically significant in elderly women.[21] Another study of suicides during SARS suggested that disconnectedness and fear of contracting SARS were more prevalent in older adult SARS-related suicide victims than non-SARS-related suicide victims. However, this was based on a small number of suicides.[23] To summarise, at present, our understanding of the effects of pandemics on suicide rates is very limited.

The availability of high-quality historical data on mortality due to influenza and suicide in Sweden provides a unique opportunity to examine this important question. While we at present have some short-term data on the rates of suicide during the COVID-19 pandemic, a better understanding of what may be the outcome in the longer run would be valuable. Using historical data from the three major influenza pandemics of the 20th century, we aimed to formally investigate whether an association between influenza deaths and suicide deaths exists. We hypothesise that such association, if any, would likely be weak. Given the substantial sex differences in suicidal behaviour and the studies suggesting sex-specific effects during

SARS,[21] we additionally report on associations separately in men and women.

## METHODS
### Data acquisition
Annual data on influenza death rates and suicide rates were estimated based on information from the Statistical Yearbook of Sweden from 1910 to 1978.[24] Over this period, Sweden experienced three influenza pandemics that occurred during different sociopolitical contexts, namely, the Spanish influenza (1918–1920, with the first case appearing in Sweden in June 1918), the Asian influenza (1957–1958, being first documented in August 1957) and the Hong Kong influenza (1968–1969, starting in the autumn of 1968).[20] The yearbooks were produced by Statistics Sweden, a governmental agency responsible for the official statistics in Sweden, with a history of population statistics going back to the 18th century.[24] For each year from 1910 to 1978, we retrieved information on the total population of Sweden, the number of deaths by influenza (if death cause was indicated as 'influenza') and the number of suicides (if death cause was indicated as 'suicide'), as well as the corresponding data separately for men and women. Since yearbooks reported data retrospectively for several years prior to the year each book was published, we checked the correctness of retrieved data by comparing yearbooks with overlapping reporting periods. We constructed influenza mortality rates and suicide rates per 100 000 inhabitants for each year by dividing the number of deaths by influenza and, separately, suicide, by the total number of individuals registered in Sweden in a corresponding year and then multiplying by 100 000. The corresponding rates for men and women were constructed likewise. In addition, we collected information on the changes in registration of deaths in Sweden that included the cause of death classification based on the Bertillon criteria (prior to 1931), the new classification introduced in cooperation with statistical authorities from other Nordic countries (1931–1950), the International Classification of Diseases (ICD) Sixth Revision (ICD-6; 1951–1957), ICD-7 (1958–1968) and ICD-8 (1969–1978).[25] To capture a potential effect of changes in classification, we created a series of dummy variables for each corresponding period, but these were only kept in the models if statistically significant.

The study protocol was not preregistered. The full dataset used in the analysis of historical data (1910–1978) and the STATA code are available in the online supplemental eTable 1.

### Statistical analyses
We used the historical data 1910–1978 that include three pandemics and focused on exploring a possible asymmetric short-term effect of influenza death rates on suicide (ie, immediate or instantaneous effect) and long-term effect (ie, if the relationship of exposure and outcome has a lag-structure, or in other words, if the

effect of exposure on outcome is distributed over a longer period of time) by applying non-linear autoregressive distributed lag (NARDL) models,[26] a technique initially introduced for research in economics. NARDL models split the exposure in partial sum of positive changes (ie, increases in influenza death rates) and partial sum of negative changes (ie, decreases in influenza death rates) and explore if an outcome responds differently (ie, asymmetrically) to an increase and decrease in exposure variable.[26] In other words, NARDL models do not rely on an assumption of symmetrical effects, by which the association of the outcome with a unit of positive change in the exposure is expected to be equal in strength and opposite in direction to the association between the outcome and a unit of negative change in the exposure. We performed the modelling by using the *nardl* command in STATA.

Before the NARDL model is executed, it is important to test that the conditions for the modelling are fulfilled. It starts from testing whether the variables are stationary (ie, their means and variances are constant over time). The advantage of the NARDL modelling is that it can be used regardless of whether the variables are integrated of the order zero, which means that variables are stationary, integrated of the order one, which indicates non-stationarity, or mixed.[26] We examined the stationarity properties of the variables by using the Augmented Dickey-Fuller unit root test (ADF) and Kwiatkowski-Phillips-Schmidt-Shin test for stationarity (KPSS) to explore the order of the integration for influenza death rates and suicide rates separately for the total population, men and women. Previous studies on suicides hypothesised that various suicidogenic factors may interplay and reinforce each other, and the logarithmic transformation of suicide time series was suggested.[27 28] We made such transformation by expressing suicide rates in natural logarithm. To select the optimal number of lags (ie, how far in the past the dependency among measurements is examined) to be used in the NARDL for dependent and independent variables, we applied the *varsoc* command in STATA using the minimal values of Akaike Information Criterion, Schwarz's Bayesian information criterion (SBIC) and the Hannan and Quinn information criterion information criteria. If information criteria indicated different lag orders, SBIC was used to select optimal lags. We applied the NARDL model to: (1) estimate the coefficients and the corresponding 95% CIs for short-term and long-term association of suicide rates with an increase and decrease in influenza death rates; (2) explore if there is a long-term cointegration between exposure and outcome by using a bounds test (if variables are cointegrated, it means their positive and negative components do not drift far away from each other in the longer term); and (3) obtain Wald test statistics that specifies whether short-term and/or long-term relation between the exposure and outcome is asymmetrical. Model diagnostic tests are described in the online supplemental material.

The modelling was performed, first, for the whole period of 1910–1978, and then for the periods of 1918–1956

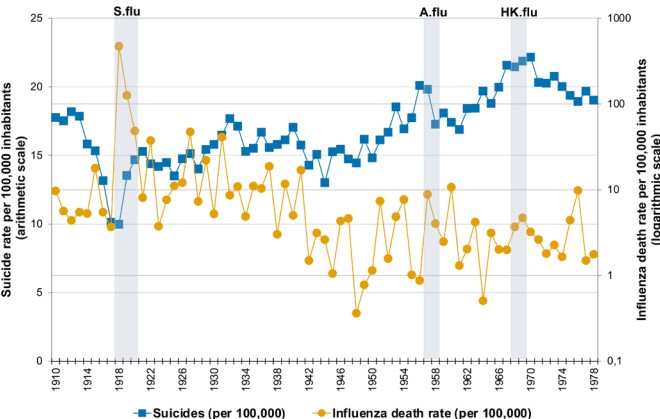

**Figure 1** Annual suicide rate (scaled on the left arithmetic y-axis) and influenza mortality (scaled on the right logarithmic y-axis) in Sweden in 1910–1978 per 100 000 inhabitants. Note: to ease visualisation, influenza death rates are reported on the logarithmic scale, while for the analysis, the logarithmic transformation was performed for suicide rates. Grey areas denote the time periods corresponding to the Spanish influenza pandemic ('S.flu', 1918–1920), the Asian influenza pandemic ('A.flu', 1957–1958) and the Hong Kong influenza pandemic ('HK.flu', 1968–1969) in Sweden.

(from the beginning of the Spanish influenza pandemic to the year before the Asian influenza pandemic) and for 1957–1978 (from the beginning of Asian pandemic to the end of observation period, with this interval also covering the Hong Kong influenza pandemic).

All statistical analyses were performed using STATA V.15.1 (StataCorp LLC).

## Patient and public involvement
Patients and members of the public were not directly involved as this study used publicly available historical national mortality data.

## RESULTS
Over the period 1910–1978, influenza death rates fluctuated considerably in Sweden with the highest rates being observed during the Spanish influenza (in 1918, 1919 and 1920, the rates were 470.93, 125.55 and 48.32 per 100 000 inhabitants, respectively) (figure 1). In postpandemic years, several noticeable peaks in influenza death rates appeared, with the ones in 1922, 1927, 1929, 1931, 1937 and 1941 being particularly high. In the following years, a considerable fluctuation in influenza mortality remained, although the rates during the periods of the Asian influenza (in 1957 and 1958: 8.78 and 3.99 per 100 000 inhabitants, respectively) and the Hong Kong influenza (in 1968 and 1969: 3.67 and 4.68 per 100 000 inhabitants, respectively) were lower than that in the first half of the century. The influenza death rates were very similar in men and women across the entire observation period (figure 2).

During the same period, a total of 80 058 deaths due to suicide occurred in Sweden (60 713 in men and 19 345 in

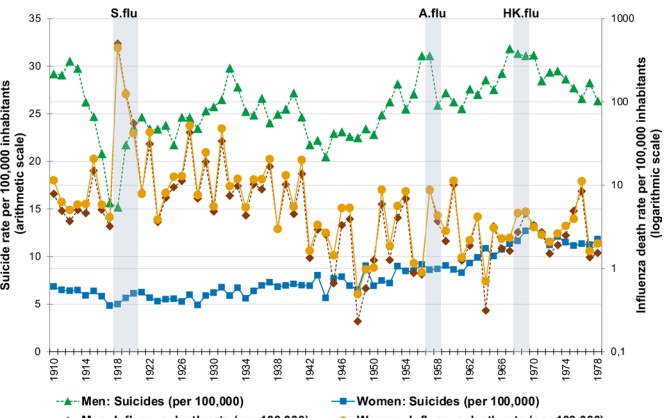

**Figure 2** Sex-specific annual suicide rate (scaled on the left arithmetic y-axis) and influenza mortality (scaled on the right logarithmic y-axis) in Sweden in 1910–1978 per 100 000 inhabitants of corresponding sex. Note: to ease visualisation, influenza death rates are reported on the logarithmic scale, while for the analysis, the logarithmic transformation was performed for suicide rates. Grey areas denote the time periods corresponding to the Spanish influenza pandemic ('S.flu', 1918–1920), the Asian influenza pandemic ('A.flu', 1957–1958) and the Hong Kong influenza pandemic ('HK.flu', 1968–1969) in Sweden.

women). The average suicide rate across 1910–1978 was 16.79 per 100 000 inhabitants (SD of 2.58), with corresponding rates for men and women as 25.85 (SD=3.37) and 7.91 (SD=2.26) per 100 000, respectively. There was a considerable fluctuation in the suicide rates over time (figure 1). The initial decrease in suicide rates during 1913–1918, with the lowest rate of 9.97 per 100 000 in 1918, was followed by an increase with the highest peak of 22.15 per 100 000 reached in 1970, with a series of intermediate peaks.

Sex-specific suicide rates differed considerably with the rates among men being between twice to over four times higher than that in women (on average, 3.4 times higher) (figure 2). Moreover, suicide rates in men exhibited a sharp dip in 1913–1918, which was first followed by an increase by 1921 and then continued to raise more gradually, although with several distinct peaks and, rarely, dips. Among women, an upward trend in suicide rate was present, although with less fluctuations than that in men.

In our analysis of 1910–1978 data, the first step with the use of the ADF and KPSS test statistics on variables' stationarity indicated that the logarithmically transformed suicide rates were integrated of the order one (for all, and men and women separately), whereas the influenza death rates were integrated of the order zero (for all and for men and women separately) (online supplemental eTable 2). On this premise, we implemented the NARDL modelling. Dummy variables on changes in death registration did neither appear statistically significant nor affected any parameter estimates in the initial model and thus were not included in the final models. As reported in table 1, for the observation period of 1910–1978, there were no statistically significant associations (either short term or long term) between decreased or

increased influenza death rates and suicide rates among the overall population and among men. The corresponding results for women indicated a possible short-term association whereby a decrease in influenza rates seemed to be associated with a borderline decrease in suicides; however, the findings were not supported by the Wald test for asymmetry (($Wald_{SR}$), that is, the null hypothesis of $Wald_{SR}$ that an increase and decrease in influenza death rates would symmetrically affect suicide rates was not rejected). This suggests that association was likely due to the effect of other unobserved factors that may influence suicides in women. Full specification of the models is reported in online supplemental eTable 3. Full specification also includes the results of testing for long-term cointegration between influenza mortality and suicides (as reported by F-statistics for Pesaran/Shin/Smith bounds test (F_PSS)). Long-term cointegration was not found as we were unable to reject the null hypothesis of no cointegration between variables (online supplemental eTable 3, see footnotes for details on bounds test for cointegration).

The results of the additional analyses for the periods 1918–1956 and 1957–1978 also failed to provide clear evidence of association between changes in influenza mortality rates and suicide rates in either short term or long term. However, in the analysis among women, a long-term asymmetry was suggested in both periods (the Wald test for asymmetry in the long term ($Wald_{LR}$) p=0.044 and p<0.001 in the analyses of 1918–1956 and 1957–1978, respectively) and a short-term asymmetry in the analysis of 1957–1978 ($Wald_{SR}$ test p=0.019) (tables 2 and 3). We assume that these findings may again reflect the influence of unobserved confounders, as these results were not supported by other coefficients, which correspond to long-term and short-term associations between increases and decreases in influenza deaths and suicides among women in 1918–1956 and 1957–1978 (tables 2 and 3). It is important to mention that the results for 1957–1978 should be considered with caution since the analysis included time series with only 22 observations. Full specifications for the models used in the analyses of 1918–1956 and 1957–1978 are reported in online supplemental eTables 4 and 5. No evidence of long-term cointegration between influenza and suicide rates were found in either period (F_PSS statistics does not reject the null hypothesis in the analyses of the whole population, men and women; online supplemental eTables 4 and 5, see footnotes for details on bounds test for cointegration). Overall, the model diagnostics for all models for 1910–1978, 1918–1956 and 1957–1978 time series support the validity of the results since, with very few exceptions, all diagnostic tests (for details, see online supplemental materials) are insignificant indicating that there is no autocorrelation, heteroscedasticity, misspecification and non-normality. In addition, the tests of the cumulative sum of recursive residuals and their squares (CUSUM and CUSUMQ, respectively) for all models indicate stability and absence of structural breaks (for details, see footnotes for tables 1–3 and online supplemental eTables 3–5).

**Table 1** Short-term and long-term relationship of suicide rates with positive and negative changes in influenza death rates among the whole population, men and women in 1910–1978 in Sweden

| | Whole population | | Men | | Women | |
|---|---|---|---|---|---|---|
| | Coeff. (95% CI) | P value | Coeff. (95% CI) | P value | Coeff. (95% CI) | P value |
| **Short-term coefficients** | | | | | | |
| Influenza + | 0.00002 (−0.00036 to 0.00039) | 0.931 | 0.00004 (−0.00034 to 0.00041) | 0.854 | −0.00007 (−0.00056 to 0.00041) | 0.760 |
| Influenza − | 0.00103 (−0.00579 to 0.00785) | 0.764 | −0.00192 (−0.01005 to 0.00621) | 0.638 | 0.00780 (0.00015 to 0.01544) | 0.046 |
| **Long-term coefficients** | | | | | | |
| Influenza + | −0.00012 | 0.998 | −0.01314 | 0.745 | 0.11254 | 0.538 |
| Influenza − | 0.00211 | 0.962 | 0.01443 | 0.722 | −0.10789 | 0.544 |
| **Model diagnostics** | | | | | | |
| Q-test for autocorrelation, $\chi^2$ | 40.160 | 0.125 | 35.260 | 0.273 | 31.850 | 0.424 |
| Heteroscedasticity, $\chi^2$ | 3.649 | 0.056 | 5.439 | 0.020 | 1.280 | 0.258 |
| Normality, $\chi^2$ | 2.826 | 0.243 | 2.237 | 0.327 | 2.372 | 0.305 |
| RESET, F-statistics | 4.656 | 0.006 | 5.473 | 0.023 | 1.013 | 0.394 |
| CUSUM | Stabl., no str. break | NA | Stabl., no str. break | NA | Stabl., no str. break | NA |
| CUSUMQ | Stabl. | NA | Stabl. | NA | Stabl. | NA |
| Adj. $R^2$ | 0.257 | NA | 0.300 | NA | 0.374 | NA |
| $Wald_{SR}$, F-statistics | 0.054 | 0.817 | 0.275 | 0.602 | 1.062 | 0.307 |
| $Wald_{LR}$, F-statistics | 1.206 | 0.277 | 1.176 | 0.283 | 0.771 | 0.384 |

Note: signs as '+' and '−' denote the exposure variable being partitioned in positive and negative changes, respectively. Wald test for asymmetry in the short term ($Wald_{SR}$) and long term ($Wald_{LR}$) has a null hypothesis of no cointegration. Q-test for autocorrelation reports the results of Portmanteau test for white noise. Heteroscedasticity is measured by Breusch-Pagan/Cook-Weisberg test. Normality is measured by Jarque-Bera test. RESET statistics refers to the regression specification error tests. CUSUM and CUSUMQ refer to the tests of the cumulative sum of recursive residuals and their squares, respectively. 'NA' denotes that a certain test parameter was not applicable. Model diagnostics support the validity of the results since, with very few exceptions (RESET statistics for the whole population and men, and heteroscedasticity for men), all diagnostic tests are insignificant indicating that there is no autocorrelation, heteroscedasticity, misspecification and non-normality. In addition, all CUSUM and CUSUMQ tests all models indicate stability and absence of structural breaks ('Stabl., no str. break' in the results of CUSUM indicates that the model was found stable, and the null hypothesis of no structural break was not rejected (same applies to 'Stabl.' as an output for CUSUMQ)).

## DISCUSSION

This study used publicly available Swedish national data from 1910 to 1978 to shed light on the potential association between influenza-related deaths and over 80 000 deaths by suicide across the three 20th century pandemics. To our knowledge, this is the first study of influenza death and suicide that analyses data from several influenza pandemics. The full modelling provided no clear evidence of either a short-term or a long-term relationship between changes in influenza death rates and changes in suicide rates. The year with the highest number of influenza-related deaths by far, 1918, had the lowest number of suicides in the whole time series.

### The findings in context

International historical data on pandemics and suicide are very scarce. A report from the USA focusing on the years 1910–1920 and using monthly data suggested a possible association between influenza deaths and suicide, but because the data did not go beyond the last year of that pandemic, it does not inform us about any longer term effects.[17] Strengths of that study included the use of monthly data and the use of time series analysis, rather than just comparing suicide rates before and after the pandemic. Chang *et al* studied suicides during the same pandemic in Taiwan, not using a time series analysis but also using monthly data from 1918 to 1920 and found an increase during the few months of the second vawe of the pandemic but that the effect was short lived. As our study used yearly and not monthly data, it could not confirm or disconfirm the findings of the Taiwanese study. Our study expands and improves on previous investigations because it allowed us to examine the long-term associations between influenza and suicide deaths. A study focusing on SARS and suicide found some evidence of sex-specific effects in the short term.[21] As suicide rates varied considerably in the years preceding SARS in Hong Kong, it is difficult to be certain that SARS, and not other contributing factors, was causally associated with an increase in suicides. In the present study, which uniquely spanned over several decades, we found that while suicide was consistently more prevalent among men throuought the study period, there were no sex-specific associations to influenza death.

**Table 2** Short-term and long-term relationship of suicide rates with positive and negative changes in influenza death rates among the whole population, men and women in 1918–1956 in Sweden

| | Whole population | | Men | | Women | |
|---|---|---|---|---|---|---|
| | Coeff. (95% CI) | P value | Coeff. (95% CI) | P value | Coeff. (95% CI) | P value |
| **Short-term coefficients** | | | | | | |
| Influenza + | 0.00002 (−0.00077 to 0.00082) | 0.955 | −0.00014 (−0.00102 to 0.00075) | 0.755 | 0.00098 (−0.00068 to 0.00263) | 0.237 |
| Influenza − | 0.00087 (−0.00643 to 0.00817) | 0.809 | −0.00132 (−0.01001 to 0.00741) | 0.760 | 0.00434 (−0.00595 to 0.01463) | 0.394 |
| **Long-term coefficients** | | | | | | |
| Influenza + | 0.00011 | 0.994 | −0.00490 | 0.802 | 0.01403 | 0.521 |
| Influenza − | 0.00103 | 0.936 | 0.00548 | 0.775 | −0.01098 | 0.600 |
| **Model diagnostics** | | | | | | |
| Q-test for autocorrelation, $\chi^2$ | 14.130 | 0.658 | 9.674 | 0.917 | 9.614 | 0.919 |
| Heteroscedasticity, $\chi^2$ | 0.939 | 0.333 | 0.768 | 0.381 | 5.042 | 0.025 |
| Normality, $\chi^2$ | 0.420 | 0.811 | 0.132 | 0.936 | 2.383 | 0.304 |
| RESET, F-statistics | 0.413 | 0.745 | 0.764 | 0.524 | 0.505 | 0.683 |
| CUSUM | Stabl., no str. break | NA | Stabl., no str. break | NA | Stabl., no str. break | NA |
| CUSUMQ | Stabl. | NA | Stabl. | NA | Stabl. | NA |
| Adj. $R^2$ | 0.309 | NA | 0.345 | NA | 0.432 | NA |
| Wald$_{SR}$, F-statistics | 0.056 | 0.815 | 0.155 | 0.697 | 0.183 | 0.673 |
| Wald$_{LR}$, F-statistics | 1.555 | 0.222 | 0.217 | 0.645 | 4.479 | 0.044 |

Note: signs as '+' and '−' denote the exposure variable being partitioned in positive and negative changes, respectively. Wald test for asymmetry in the short term (Wald$_{SR}$) and long term (Wald$_{LR}$) has a null hypothesis of no cointegration. Q-test for autocorrelation reports the results of Portmanteau test for white noise. Heteroscedasticity is measured by Breusch-Pagan/Cook-Weisberg test. Normality is measured by Jarque-Bera test. RESET statistics refers to the regression specification error tests. CUSUM and CUSUMQ refer to the tests of the cumulative sum of recursive residuals and their squares, respectively. 'NA' denotes that a certain test parameter was not applicable. Model diagnostics support the validity of the results since, with one exception (heteroscedasticity for women), all diagnostic tests are insignificant indicating that there is no autocorrelation, heteroscedasticity, misspecification and non-normality. In addition, all CUSUM and CUSUMQ tests all models indicate stability and absence of structural breaks ('Stabl., no str. break' in the results of CUSUM indicates that the model was found stable, and the null hypothesis of no structural break was not rejected (same applies to 'Stabl.' as an output for CUSUMQ)).

### Meaning of the study

What do these results mean for our understanding of the short-term and long-term consequences of influenza pandemics? Our results do not support the belief that global pandemics necessarily lead to a substantial increase in suicides. There are many factors associated with a global pandemic that may potentially increase suicide risk in the population, but these may be offset by other protective factors that should not be overlooked. A shared sense of belonging and focus, social connectedness and a 'pulling together effect' may be one such factor.[1 29] We acknowledge that these historical findings may not be directly applicable to the current COVID-19 pandemic because there are several important differences between the previous and current pandemics, such as globalisation, mortality rates, a different sociopolitical context and the impact of social media, to name a few.

### Strengths and limitations

The major strengths of this study were the use of a large amount of historical national data across three 20th century pandemics and state of the art time-series

analyses. A limitation is that all data come from a single country, and thus caution is needed when generalising the study findings to countries with different level of development of clinical and preventive medicine, social support, etc. The study used the Swedish historical public death records, and we had no means of verifying the causes of death. The coverage and precision of these records is likely to have improved over time for both variables. To guard against effects of changes in the recording of causes of death, we created a series of dummy variables for each corresponding period but no significant effects of those were found. If there were some other time-varying factors that could affect the coverage and precision of death records, apart from the official changes in registration system, this might have biased our results, in particular if such factors differentially affected the quality of recording deaths due to influenza and suicide. As we did not have access to data with higher temporal resolution than yearly data, it could affect our results if the time sequence of association between changes in influenza death and suicide differ from the chosen time interval.

**Table 3** Short-term and long-term relationship of suicide rates with positive and negative changes in influenza death rates among the whole population, men and women in 1957–1978 in Sweden

| | Whole population | | Men | | Women | |
|---|---|---|---|---|---|---|
| | Coeff. (95% CI) | P value | Coeff. (95% CI) | P value | Coeff. (95% CI) | P value |
| **Short-term coefficients** | | | | | | |
| Influenza + | −0.00964 (−0.02252 to 0.00324) | 0.883 | −0.00876 (−0.02738 to 0.00985) | 0.328 | −0.00976 (−0.02033 to 0.00081) | 0.068 |
| Influenza − | 0.00194 (−0.02614 to 0.03003) | 0.130 | −0.01531 (−0.04658 to 0.01596) | 0.309 | 0.02155 (−0.00343 to 0.04653) | 0.085 |
| **Long-term coefficients** | | | | | | |
| Influenza + | 0.01255 | 0.618 | −0.02659 | 0.541 | 0.04091 | 0.064 |
| Influenza − | −0.01033 | 0.684 | 0.024538 | 0.582 | −0.02870 | 0.184 |
| **Model diagnostics** | | | | | | |
| Q-test for autocorrelation, $\chi^2$ | 5.662 | 0.773 | 3.790 | 0.925 | 5.243 | 0.813 |
| Heteroscedasticity, $\chi^2$ | 0.102 | 0.750 | 0.155 | 0.694 | 0.079 | 0.778 |
| Normality, $\chi^2$ | 1.569 | 0.456 | 1.627 | 0.443 | 1.237 | 0.537 |
| RESET, F-statistics | 5.839 | 0.014 | 3.348 | 0.064 | 1.089 | 0.398 |
| CUSUM | Stabl., no str. break | NA | Stabl., no str. break | NA | Stabl., no str. break | NA |
| CUSUMQ | Stabl. | NA | Stabl. | NA | Stabl. | NA |
| Adj. $R^2$ | 0.246 | NA | 0.337 | NA | 0.371 | NA |
| Wald$_{SR}$, F-statistics | 2.903 | 0.112 | 0.536 | 0.477 | 7.132 | 0.019 |
| Wald$_{LR}$, F-statistics | 0.922 | 0.354 | 0.297 | 0.595 | 26.74 | <0.001 |

Note: signs as '+' and '−' denote the exposure variable being partitioned in positive and negative changes, respectively. Wald test for asymmetry in the short term (Wald$_{SR}$) and long term (Wald$_{LR}$) has a null hypothesis of no cointegration. Q-test for autocorrelation reports the results of Portmanteau test for white noise. Heteroscedasticity is measured by Breusch-Pagan/Cook-Weisberg test. Normality is measured by Jarque-Bera test. RESET statistics refers to the regression specification error tests. CUSUM and CUSUMQ refer to the tests of the cumulative sum of recursive residuals and their squares, respectively. 'NA' denotes that a certain test parameter was not applicable. Model diagnostics support the validity of the results since, with one exception (RESET statistics for the whole population), all diagnostic tests are insignificant indicating that there is no autocorrelation, heteroscedasticity, misspecification and non-normality. In addition, all CUSUM and CUSUMQ tests all models indicate stability and absence of structural breaks ('Stabl., no str. break' in the results of CUSUM indicates that the model was found stable, and the null hypothesis of no structural break was not rejected (same applies to 'Stabl.' as an output for CUSUMQ)).

A multitude of factors may vary that impact the resilience of the society with regards to the effect of a pandemic. Such factors may also vary over time and place. However, the fact that we observed no clear associations between influenza and suicide deaths across pandemics, which challenged society with various degrees of, for example, economic effects and healthcare supply issues, does support the stability of our findings.

## CONCLUSION

In this national analysis of historical data spanning across three 20th century pandemics, we found no evidence of a short-term or long-term association between influenza death rates and suicide rates. The results challenge the notion that an increase of suicides follows as a certain consequence of a pandemic. Future research on the effect of the current COVID-19 pandemic should explore the possibility of differential short-term and long-term effects on suicide rates in different subgroups of the population.

**Contributors** CR and AS have full access to all the data in this study and take full responsibility as guarantors for the integrity of the data and the accuracy of the data analysis. CR, DM-C and AS conceived and designed the study. CR, KM, MA and AS performed data collection. AS undertook the statistical analysis. CR, DM-C and AS drafted the manuscript. All authors provided critical input to the analyses, interpreted the data and revised the manuscript critically. The corresponding author confirms that all listed authors meet authorship criteria and that no others meeting the criteria have been omitted.

**Funding** Karolinska Institutet and Region Stockholm, which provide salary for the authors, had no involvement in the study design; collection, analysis and interpretation of data; in the writing of the report; or in the decision to submit the paper for publication.

**Competing interests** None declared.

**Patient consent for publication** Not required.

**Provenance and peer review** Not commissioned; externally peer reviewed.

**Data availability statement** All data relevant to the study are included in the article or uploaded as supplementary information. The full dataset (numbers of deaths by influenza and suicide and total population for 1910-1978) is published as supplementary material (online supplemental eTable 1). The STATA code used in the analyses is available in the online supplement.

responsibility arising from any reliance placed on the content. Where the content includes any translated material, BMJ does not warrant the accuracy and reliability of the translations (including but not limited to local regulations, clinical guidelines, terminology, drug names and drug dosages), and is not responsible for any error and/or omissions arising from translation and adaptation or otherwise.

**ORCID iDs**
Christian Rück http://orcid.org/0000-0002-8742-0168
Oskar Flygare http://orcid.org/0000-0002-2017-3940

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
