## [Reviewer comments · BMJ Open]

ARTICLE DETAILS

TITLE (PROVISIONAL)	A Swedish nationwide time series analysis of influenza and suicide deaths from 1910 to 1978
AUTHORS	Rück, Christian; Mataix-Cols, David; Malki, Kinda; Adler, Mats; Flygare, Oskar; Runeson, Bo; Sidorchuk, Anna

VERSION 1 – REVIEW

REVIEWER	Iorfa, Steven University of Nigeria, Psychology
REVIEW RETURNED	17-Feb-2021

GENERAL COMMENTS	This is an excellent study. However, I feel it can be greatly improved if the authors clearly state out their objectives and define their outcome. From the topic, it seems as though they are investigating the link between pandemics and suicides. However, in the work, they seem to check for the association between deaths during pandemics and suicides during pandemics. A better approach would have been to investigate the prevalence of suicides during pandemics against times when there were no pandemics. If not, I suggest they reframe their title.
--

REVIEWER	Okusaga, Olaoluwa Baylor College of Medicine, Psychiatry and Behavioral Sciences
REVIEW RETURNED	18-Feb-2021

GENERAL COMMENTS	The authors have addressed an important question: do pandemics result in increased suicide rates? They made use of Swedish national data from 1910 to 1978 and employed a statistical approach not frequently used in the health and biological sciences. Their results, counterintuitively, did not confirm an association between pandemics and increased deaths from suicide. However, the variable "influenza deaths" was used as an indicator of a pandemic; therefore, the question that remains unanswered is whether deaths from influenza (or any other infectious agent) is the best indicator of a pandemic. Simply put, is an influenza pandemic=number of deaths from influenza? If influenza pandemic is not just simply the number of deaths from influenza, then the authors should address this issue in the manuscript and should also consider conducting additional analyses in which suicide rates during the periods of the three pandemics (1918-1920, 1957-1958 and 1968-1969) are compared with suicide rates during periods of no pandemic and see if there are significant differences or not. The authors actually used this approach when they compared suicide rates for the first six months in 2019 with suicide rates in the first six months of 2020 (i.e. rates during a pandemic and rates during a period of no pandemic).
---

	In addition, the authors wrote "we suggest a more moderated message to the public to avoid unfounded fear and an ineffective use of resources for prevention and care". I do not think that this one study based on data from one country should necessarily change practice or the messaging when it comes to suicide prevention during a pandemic such as ongoing COVID-19 pandemic. As a matter of fact, the authors should acknowledge that a limitation of their study is that it is based on data from one country and therefore caution should be exercised in generalizing the study findings to other countries.
--	---

REVIEWER	Leske, Stuart Griffith University
REVIEW RETURNED	22-Mar-2021

GENERAL COMMENTS	I commend the authors on this important and, to my knowledge, unique study, which could provide important insights into the trajectories of suicides in Sweden and other countries in this pandemic. My comments are mostly minor. Page 2, line 48: I would add 'to our knowledge,' before 'First study,' to be safe. Page 4, lines 18-21: With respect to the US analysis, I would encourage the authors to incorporate references to the work of Sam Harper, who has found contrasting results with those earlier studies after accounting for seasonality and trends: https://pubmed.ncbi.nlm.nih.gov/28625812/ Page 4, lines 18-23: There's also some studies of Spain (https://www.sciencedirect.com/science/article/pii/S2173505017300237) and Ireland (https://academic.oup.com/ije/article/44/3/969/631832) Page 4, lines 36-38: While not as relevant, there is a qualitative study looking at mechanisms for increases in older adult suicides in SARS: https://pubmed.ncbi.nlm.nih.gov/20418214/ Page 3, line 3, page 4, line 29; 38: Need British English spelling here (summarise, standardise). Page 6, lines 23-24: Could the authors put confidence intervals around these rates instead of SDs? Page 8, lines 22-23: Could you please spell out 'F_PSS' the first time you mention it? Page 8, lines 27-43: While there is considerable methodological detail accompanying the historical analyses, the paragraph detailing the analysis of current data stands in quite stark contrast to this. I appreciate that there are likely genuine reasons (e.g., power issues due to limited post-exposure time points) why the authors adopted a descriptive approach rather than an interrupted time-series approach here. The problem is that I think that's what readers would be expecting for this analysis, and this is the pandemic that most people are interested in. For some justification of why you might not do an ITS here, please see the 'data requirements' section of Bernal et al., 2017: https://academic.oup.com/ije/article/46/1/348/2622842 and note the power calculation article by Zhang: https://pubmed.ncbi.nlm.nih.gov/21640554/. Even so, at least an ITS
--

	would tell you whether there is a departure from the expected post-exposure values based on the pre-exposure trend. Presumably, lower power would be reflected in wider CIs, although the direction of the RR would still be informative. Anyway, I note to the authors that the emphasis on the statistical analyses of the historical datasets is disproportionate to the current analysis, so this requires some justification for why this is so. Perhaps, because the data is of high quality, the historical pandemics are of most interest in telling us what we should expect in this pandemic? Page 9, line 9: 'causally associated' sounds like an oxymoron. Could you omit 'was causally associated with an increase in suicides' and say 'increased suicides'? Page 8, lines 26-28: I would say these are multiple strengths, so I would change 'is' to 'are.' Page 8, lines 56-58: I appreciate that the authors probably never thought the pandemic would lead to a tsunami of suicides, but when I first saw this word in the preprint of the paper posed as a leading question, I found the language surprising (I acknowledge the submitted version has a different title). I think you could instead finish on the note summarising your analysis, that you've looked at long-term data after three pandemics and found no increases in suicides. Therefore, based on the impact of past pandemics in Sweden, we should not expect an increase in this pandemic. The general media reporting of suicides itself confers only a small increased rate ratio, but reporting celebrity suicides would be problematic: https://www.bmj.com/content/368/bmj.m575. I'm also not sure how a more moderated message would avoid an ineffective use of resources for prevention and care. Perhaps because there would be more considered dissemination of funding? I think that all gets away from your very important and unique findings, so I would encourage you to end on those. Page 14-15: For the tables, it would be helpful if you could highlight in bold (or whatever BMJ Open requests) any p-values the reader should take notice of as indicating a violation of an assumption of the model. That way, we can reconcile this assumption violation in the table with any description of steps taken to address it in the text. Perhaps there were no assumption violations. Figures 1-3: I appreciate the authors have taken steps to ease interpretation in the figures. I still think they are too busy and inconsistent with recommendations. I would therefore ask the authors to make them all consistent per: https://onlinelibrary.wiley.com/doi/10.1002/jrsm.1435 (they supply the Stata code, which will work in the version of Stata you have). You won't need to plot counterfactuals, and I think the grey shading for exposures that you have is much better than a red line (which doesn't convey the period). But I think the colours as they stand are too inconsistent and not easy on the eye. I would also suggest reducing the years listed on the x-axis. Maybe every 4 or 5 years. Lastly, I could not see STROBE and RECORD checklists accompanying the material, so the article would be strengthened by these populated checklists (some items in them may be n/a).
--	--

REVIEWER	Bray, Michael Johns Hopkins Medicine, Institute of Medical Science
----------	---

GENERAL COMMENTS

The authors are to be commended on this interesting manuscript. This article has several notable strengths, including utilization of nation-wide suicide data over multiple years (including during the COVID-19 pandemic), potentially meaningful subgroup analyses, and the timeliness of the findings. With this in mind, I have several specific suggestions and comments, which, if integrated/addressed, could constitute a meaningful improvement to this addition to the literature:

The introduction would benefit from a more thorough review of the current literature surrounding COVID-19 and suicide mortality. Undoubtedly, the authors' review was conducted prior to the publication of some of these articles but it is essential that their findings be placed in the context of these other key findings. For instance, current data is in line with the notion that suicide mortality is not in fact increasing on the whole during the COVID-19 pandemic but this is now supported by more concrete findings regarding suicide mortality than the authors cite (see Faust et al 2021 JAMA Network Open; Appleby et al 2021 preprint; Bray et al 2020 JAMA Psychiatry; Mitchell et al 2021 Psychiatr Res; Leske et al 2021 Lancet Psychiatry). Furthermore, the authors commendable approach using subgroup analyses is supported by emerging data that some subgroups bear differing burden of suicide mortality during the COVID-19 pandemic, though the current literature has found increased suicide mortality among African Americans as opposed to gender differences (see Bray et al 2020 JAMA Psychiatry and Mitchell et al 2021 Psychiatr Res). These subgroup differences warrant comment in the introduction. A minor comment, but the authors state in the introduction that the Spanish flu was accompanied by an increase in suicide – this is confounded by the end of World War I and warrants mention. I might recommend softer wording (for instance “the Spanish flu coincided with an increase in suicides” as opposed to “was associated with”).

In the introduction, more context is needed in order to quickly characterize socioeconomic fallout of other pandemics. If we are to extrapolate any of these findings to the current COVID-19 pandemic I am interested in knowing more about lockdowns, joblessness, etc. during the Spanish flu and others in Sweden. The authors report on suicides separately in men and women, while this is to be commended, your hypotheses related to the intersection of gender, pandemics and suicide are unclear. Do you expect men and women to respond differently to the stresses of the pandemic regarding suicide risk? What literature supports this? At the end of the introduction, please include any directional hypotheses that were made a priori to strengthen the impact of these findings further.

The dummy variable accounting for different coding of outcomes is very well thought out.

One limitation of the authors' statistical approach is that it appears to operationalize effects of influenza deaths on suicide mortality as consistent from 1910 to 2020, without accounting for the possibility that influenza deaths during pandemics may be starkly different from those suffered outside of the context of pandemics. As such, it is difficult to draw the conclusion that suicides are not affected by pandemics based on these analyses. The authors should address this in the revised manuscript.

It seems to me that, during the Spanish flu, suicide rates trend upwards for the duration of the pandemic (with the exception of the

	first year which marks the end of World War I). I'm under the impression that this was the only of the three pandemics studied that was defined by widespread sociological changes such as lockdowns. In light of this, the degree to which we can extrapolate these results to the current pandemic is unclear, though I would appreciate the authors weighing in on this in a revised manuscript. How was "short term" and "long term" operationalized? Greater clarification of this is warranted in the methods section. When was the start of the COVID-19 pandemic in Sweden? (i.e. first case, first death, first lockdowns, etc.) It seems to me that including January in analyses as a "COVID month" may be watering down the findings by including irrelevant months if it hit in earnest during February. A description of this should be included for readers unfamiliar with the timeline of Sweden's COVID-19 course. Further, rationale for inclusion of January would be helpful. In results, the authors state that significant Wald tests may be discounted as these results "are not supported by other coefficients". While I agree that these Wald tests are not strong evidence of a relationship, explicitly stating which coefficients are not consistent might be helpful. There seems to be no statistical testing in the comparison of 2020 and 2019 suicide rates so it is challenging to evaluate as a reader if these are meaningful difference. Was data only available in aggregate? As stated above regarding the introduction, a more thorough review of the latest literature on COVID and suicide mortality should be considered to properly place these findings in the appropriate context. This would be helpful to demonstrate that these findings are consistent with this growing literature suggesting that suicides are not increasing in the aggregate. In the discussion the authors mention that 1918 had the lowest suicide mortality despite highest influenza mortality. This is almost certainly confounded by the end of WWI and if this is not presented in this context it may be misleading for some readers. The final sentence of the conclusion is quite strong, particularly on the basis of these data. A more moderate message is certainly reasonable in light of the growing literature suggesting suicide rates are not increasing in aggregate (though some sub-groups may be at elevated risk), though this conclusion is best made in the light of this growing body of literature. Dispelling moral panic at this stage is reasonable, though I would caution against advocating against allocation of resources for suicide prevention and mental health care. I greatly appreciated the opportunity to review this interesting manuscript. Again, the authors are to be commended for taking on this potentially impactful project. I am hopeful that their consideration of the above comments will be helpful to the authors as they revise this manuscript further and I am optimistic it will constitute a meaningful contribution to the literature.
--	---

VERSION 1 – AUTHOR RESPONSE

Reviewer: 1

Dr. Steven Iorfa, University of Nigeria

Comments to the Author:

This is an excellent study. However, I feel it can be greatly improved if the authors clearly state out their objectives and define their outcome. From the topic, it seems as though they are investigating the link between pandemics and suicides. However, in the work, they seem to check for the association between deaths during pandemics and suicides during pandemics. A better approach would have been to investigate the prevalence of suicides during pandemics against times when there were no pandemics. If not, I suggest they reframe their title.

Reply: Thank you for kind words. The title has been reframed. We want to clarify that the analysis includes data from 1910 to 1978, that is, we do not only use data from the pandemic periods. In our view, in order to answer our research question, it would not be enough just to compare the prevalence of suicides during pandemics and the periods of no pandemics because the relationship between exposure (influenza rates) and outcomes (suicide rates) might have a lag-structure, i.e., to be distributed over a longer period of time. Moreover, the relationship between exposure and outcomes may have different and even opposite directions if measured instantaneously (i.e., if short-term effect is measured) and in dynamic (i.e., if long-term effect is measured). The statistical approach that we used - non-linear autoregressive distributed lag (NARDL) models – allows studying both short- and long-term effects and, most importantly, clarifies the issue of relationship between the outcome and the period when exposure increases and decreases. This additional focus on the potential different effect of increases and decreases of exposure on outcome helps to clarify the complexity of exposure-outcome relationship. In light of abovementioned, we do believe that the time-series analysis, in general, and the statistical approach used (the NARDL) are superior to for instance comparing averages before and after a pandemic.

Reviewer: 2

Dr. Olaoluwa Okusaga, Baylor College of Medicine, Michael E DeBakey VA Medical Center
Comments to the Author:

The authors have addressed an important question: do pandemics result in increased suicide rates? They made use of Swedish national data from 1910 to 1978 and employed a statistical approach not frequently used in the health and biological sciences. Their results, counterintuitively, did not confirm an association between pandemics and increased deaths from suicide.

However, the variable "influenza deaths" was used as an indicator of a pandemic; therefore, the question that remains unanswered is whether deaths from influenza (or any other infectious agent) is the best indicator of a pandemic. Simply put, is an influenza pandemic=number of deaths from influenza? If influenza pandemic is not just simply the number of deaths from influenza, then the authors should address this issue in the manuscript and should also consider conducting additional analyses in which suicide rates during the periods of the three pandemics (1918-1920, 1957-1958 and 1968-1969) are compared with suicide rates during periods of no pandemic and see if there are significant differences or not. The authors actually used this approach when they compared suicide rates for the first six months in 2019 with suicide rates in the first six months of 2020 (i.e. rates during a pandemic and rates during a period of no pandemic).

Reply: Thank you for the comments. To answer them fully, we would like to split the comments into two parts: (1) on "influenza deaths" as an indicator for of a pandemic, and (2) the comparison of suicide rates during the periods of pandemics and no pandemics. For the first part of the comments - the feedback makes us realize that we need to be more precise regarding the wording when it comes to pandemic and influenza death. It is indeed true that an "influenza pandemic" might not be best presented by number of influenza deaths and the information on influenza cases (both fatal and non-

fatal) would probably be the better measure. However, the information on non-fatal cases of influenza was not available to us and the quality of death records might be considered to be superior to the quality of non-fatal records, in particular, when historical data are involved. Thus, we used the best available measure for defining influenza pandemics.

For the second part of the comments, we would like to refer to our reply that we gave on the similar comment by Reviewer 1. In our view, in order to answer our research question, it would not be enough just to compare the prevalence of suicides during pandemics and the periods of no pandemics because the relationship between exposure (influenza rates) and outcomes (suicide rates) might have a lag-structure, i.e., to be distributed over a longer period of time. Moreover, the relationship between exposure and outcomes may have different and even opposite directions if measured instantaneously (i.e., if short-term effect is measured) and in dynamic (i.e., if long-term effect is measured). The statistical approach that we used - non-linear autoregressive distributed lag (NARDL) models – allows studying both short- and long-term effects. “*Short-term effect*” refers to the immediate relation between independent time series (influenza death rates) and dependent time series (suicide rates), i.e., the relation can be established for each unit of time used (year, quarter, months, weeks, etc.). “*Long-term effect*” implies that the relationship of exposure and outcome has a lag-structure, or in other words, the effect of exposure on outcome is distributed over a longer period of time. In our article, we briefly describe the meaning of effects under the subheading “Statistical analysis” (page 4, first paragraph in the section). Most importantly, the NARDL clarifies the issue of relationship between the outcome and the period when exposure increases and decreases. This additional focus on the potential different effect of increases and decreases of exposure on outcome helps to clarify the complexity of exposure-outcome relationship. In light of abovementioned, we do believe that the time-series analysis, in general, and the statistical approach used (the NARDL) are superior to for instance comparing averages before and after a pandemic.

Also, we would like to mention that we have removed data on COVID-19. In the previous version of the manuscript, we indeed compared the suicide rates in the first six months of 2020 (the time period that refers to pandemic) with the same months in 2019. This approach was used as no other approaches were applicable for such a short data series. However, when a considerable length of time series are available (as in our historical data collection), we would still consider more sophisticated approach that allows to study a dynamic relationship – as for example, the NARDL – to be more appropriate and suitable for our study.

In addition, the authors wrote "we suggest a more moderated message to the public to avoid unfounded fear and an ineffective use of resources for prevention and care". I do not think that this one study based on data from one country should necessarily change practice or the messaging when it comes to suicide prevention during a pandemic such as ongoing COVID-19 pandemic. As a matter of fact, the authors should acknowledge that a

limitation of their study is that it is based on data from one country and therefore caution should be exercised in generalizing the study findings to other countries.

Reply: Thank you. We agree that the cited sentence, especially now that we have removed all COVID-19 data, is not ideal. We have changed it to: “The results challenge the notion that an increase of suicides follows as a certain consequence of a pandemic.” We also added one suggested limitation in the discussion on page 7: “A limitation is that all data comes from a single country and thus caution is needed when generalizing the study findings to countries with different level of development of clinical and preventive medicine, social support, etc.”

Reviewer: 3

Dr. Stuart Leske, Griffith University
Comments to the Author:

I commend the authors on this important and, to my knowledge, unique study, which could provide important insights into the trajectories of suicides in Sweden and other countries in this pandemic. My comments are mostly minor.

Reply: Thank you very much.

Page 2, line 48: I would add 'to our knowledge,' before 'First study,' to be safe.

Reply: Thank you for the suggestion, we have added that as suggested.

Page 4, lines 18-21: With respect to the US analysis, I would encourage the authors to incorporate references to the work of Sam Harper, who has found contrasting results with those earlier studies after accounting for seasonality and

trends:

<https://eur01.safelinks.protection.outlook.com/?url=https%3A%2F%2Fpubmed.ncbi.nlm.nih.gov%2F28625812%2F&data=04%7C01%7Cchristian.ruck%40ki.se%7C5815c7167262423f894108d8f9dfb2e3%7Cbff7eef1cf4b4f32be3da1dda043c05d%7C0%7C0%7C637534086465856241%7CUnknown%7CTWFpbGZsb3d8eyJWIjo%7CiMC4wLjAwMDAiLCJQIjoiV2luMzliLCJBTiI6Ikh1aWwiLCJXVCi6Mn0%3D%7C1000&data=YHAU%2FXM5b5KDbB7oRjYKWEDXVqSHegsuzij%2BB7T%2BzHU%3D&reserved=0>

Page 4, lines 18-23: There's also some studies of Spain

<https://eur01.safelinks.protection.outlook.com/?url=https%3A%2F%2Fwww.sciencedirect.com%2Fscience%2Farticle%2Fpii%2FS2173505017300237&data=04%7C01%7Cchristian.ruck%40ki.se%7C5815c7167262423f894108d8f9dfb2e3%7Cbff7eef1cf4b4f32be3da1dda043c05d%7C0%7C0%7C637534086465856241%7CUnknown%7CTWFpbGZsb3d8eyJWIjo%7CiMC4wLjAwMDAiLCJQIjoiV2luMzliLCJBTiI6Ikh1aWwiLCJXVCi6Mn0%3D%7C1000&data=hs5eQeiGKQsGujMli%2FS0EeZnJBenO9pEqsUzKMNUg2U%3D&reserved=0>) and Ireland

<https://eur01.safelinks.protection.outlook.com/?url=https%3A%2F%2Facademic.oup.com%2Fije%2Farticle%2F44%2F3%2F969%2F631832&data=04%7C01%7Cchristian.ruck%40ki.se%7C5815c7167262423f894108d8f9dfb2e3%7Cbff7eef1cf4b4f32be3da1dda043c05d%7C0%7C0%7C637534086465856241%7CUnknown%7CTWF>

[pbGZsb3d8eyJWljo iMC4wLjAwMDAiLCJQIjoiV2luMzliLCJBTiI6Ikk1haWwiLCJXVCI6Mn0%3D%7C1000&data=wPjfW6TSh2Tcl ZCQW0870ui8XsZ1mX9IWDnvyA%2FLuQ%3D&reserved=0](https://eur01.safelinks.protection.outlook.com/?url=https%3A%2F%2Fpubmed.ncbi.nlm.nih.gov%2F20418214%2F&data=04%7C01%7Cchristian.ruck%40ki.se%7C5815c7167262423f894108d8f9dfb2e3%7Cbff7eef1cf4b4f32be3da1dda043c05d%7C0%7C0%7C637534086465856241%7CUnknown%7CTWFpbGZsb3d8eyJWljo iMC4wLjAwMDAiLCJQIjoiV2luMzliLCJBTiI6Ikk1haWwiLCJXVCI6Mn0%3D%7C1000&data=wPjfW6TSh2Tcl ZCQW0870ui8XsZ1mX9IWDnvyA%2FLuQ%3D&reserved=0)

Page 4, lines 36-38: While not as relevant, there is a qualitative study looking at mechanisms for increases in older adult suicides in

SARS:

<https://eur01.safelinks.protection.outlook.com/?url=https%3A%2F%2Fpubmed.ncbi.nlm.nih.gov%2F20418214%2F&data=04%7C01%7Cchristian.ruck%40ki.se%7C5815c7167262423f894108d8f9dfb2e3%7Cbff7eef1cf4b4f32be3da1dda043c05d%7C0%7C0%7C637534086465856241%7CUnknown%7CTWFpbGZsb3d8eyJWljo iMC4wLjAwMDAiLCJQIjoiV2luMzliLCJBTiI6Ikk1haWwiLCJXVCI6Mn0%3D%7C1000&data=wPjfW6TSh2Tcl ZCQW0870ui8XsZ1mX9IWDnvyA%2FLuQ%3D&reserved=0>

Reply: Thank you for great suggestions. We have added Sam Harper's paper in the introduction and the study by Yip et al as well.

Page 3, line 3, page 4, line 29; 38: Need British English spelling here (summarise, standardise).

Reply: Thank you. This has been corrected.

Page 6, lines 23-24: Could the authors put confidence intervals around these rates instead of SDs?

Reply: Because the reported suicide rates across 1910-1978 were calculated for the whole population (or for all men and women in respective calculations), the confidence intervals (CIs) might be less informative than standard deviations (SDs) due to CIs being very narrow (this is because the sample size, i.e., the denominator, is large). Thus, the reported suicide rate of 16.79 per 100 000 inhabitants (standard deviation [SD] of 2.58) would correspond to 16.7976 (95% CI: 16,7974, 16.7978). To ease visualization of these numbers, they might be rounded up to **16.79 (95 CI: 16.79, 16.80)**. The corresponding rates and SDs for men of 25.85 (SD=3.37) per 100

000 will correspond to 25.8537 (95 CI: 25.8495, 25.8933), that might be rounded up to **25.85 (95 CI: 25.85, 25.89)**. For women, the reported rates and SD of 7.91 (SD=2.26) per 100 000 will correspond to 7.9144 (95 CI:

7.9141, 7.9147) that might be rounded up to **7.91 (95% CI: 7.91, 7.92)**. It seems to us that this might be easier for the readers to interpret the results if the average rates are reported as 'rates and SDs'. Also, this manner of reporting average rates of outcomes across time seems to be widely used in other time series analyses. However, we are ready to reconsider if the reviewer will insist on reporting rates and 95% CIs instead.

Page 8, lines 22-23: Could you please spell out 'F_PSS' the first time you mention it?

Reply: Thank you for giving us the opportunity to clarify. F_PSS is F-statistics for Pesaran/Shin/Smith bounds test. The name of the test is written in full in the Results section (page 6, paragraph 3), where the abbreviation appears for the first time.

Page 8, lines 27-43: While there is considerable methodological detail accompanying the historical analyses, the paragraph detailing the analysis of current data stands in quite stark contrast to this. I appreciate that there are likely genuine reasons (e.g., power issues due to limited post-exposure time points) why the authors adopted a descriptive approach rather than an interrupted time-series approach here. The problem is that I think that's what readers would be expecting for this analysis, and this is the pandemic that most people are interested in. For some justification of why you might not do an ITS here, please see the 'data requirements' section of Bernal et al., 2017: <https://eur01.safelinks.protection.outlook.com/?url=https%3A%2F%2Facademic.oup.com%2Fije%2Farticle%2F46%2F1%2F348%2F2622842&data=04%7C01%7Cchristian.ruck%40ki.se%7C5815c7167262423f894108d8f9dfb2e3%7Cbff7eef1cf4b4f32be3da1dda043c05d%7C0%7C0%7C637534086465861227%7CUnknown%7CTWFpbGZsb3d8eyJWljojMC4wLjAwMDAiLCJQIjoiV2luMzliLCJBTiI6I6k1haWwiLCJXVCI6Mn0%3D%7C1000&data=7p0vDKjeQD225VK4tleXwNbDarpY6uBzIkCI6a6S%2FU0%3D&reserved=0> and note the power calculation article by Zhang:

<https://eur01.safelinks.protection.outlook.com/?url=https%3A%2F%2Fpubmed.ncbi.nlm.nih.gov%2F21640554%2F&data=04%7C01%7Cchristian.ruck%40ki.se%7C5815c7167262423f894108d8f9dfb2e3%7Cbff7eef1cf4b4f32be3da1dda043c05d%7C0%7C0%7C637534086465861227%7CUnknown%7CTWFpbGZsb3d8eyJWljojMC4wLjAwMDAiLCJQIjoiV2luMzliLCJBTiI6I6k1haWwiLCJXVCI6Mn0%3D%7C1000&sdata=YGhMDyOHGx6PkXH8w3bb0wwfZQwTuvJuASr4bypwdV4%3D&reserved=0>. Even so, at least an ITS would tell you whether there is a departure from the expected post-exposure values based on the pre-exposure trend. Presumably, lower power would be reflected in wider CIs, although the direction of the RR would still be informative. Anyway, I note to the authors that the emphasis on the statistical analyses of the historical datasets is disproportionate to the current analysis, so this requires some justification for why this is so. Perhaps, because the data is of high quality, the historical pandemics are of most interest in telling us what we should expect in this pandemic?

Reply: Thank you for the detailed and helpful suggestions (we read the suggested papers with a great interest and we fully agree with the Reviewer on the limitations of our data for conducting the ITS analysis). As already mentioned, we decided to remove the COVID-19 data. We also understood from the comment, with respect to the *historical data analysis*, the Reviewer has found our statistical approach to be suitable and that we described it appropriately. We are grateful for the support given to our choice of statistical approach for analyzing historical data

Page 9, line 9: 'causally associated' sounds like an oxymoron. Could you omit 'was causally associated with an increase in suicides' and say 'increased suicides'?

Reply: Thank you for the suggestion, we have changed according to the suggestion.

Page 8, lines 26-28: I would say these are multiple strengths, so I would change 'is' to 'are.'

Reply: Since we now have removed the COVID-19 data, the sentence has been changed.

Page 8, lines 56-58: I appreciate that the authors probably never thought the pandemic would lead to a tsunami of suicides, but when I first saw this word in the preprint of the paper posed as a leading question, I found the language surprising (I acknowledge the submitted version has a different title). I think you could instead finish on the note summarising your analysis, that you've looked at long-term data after three pandemics and found no increases in suicides. Therefore, based on the impact of past pandemics in Sweden, we should not expect an increase in this pandemic. The general media reporting of suicides itself confers only a small increased rate ratio, but reporting celebrity suicides would be problematic:

<https://eur01.safelinks.protection.outlook.com/?url=https%3A%2F%2Fwww.bmj.com%2Fcontent%2F368%2Fbmj.m575&data=04%7C01%7Cchristian.ruck%40ki.se%7C5815c7167262423f894108d8f9dfb2e3%7Cbff7eef1cf4b4f32be3da1dda043c05d%7C0%7C0%7C637534086465861227%7CUnknown%7CTWFpbGZsb3d8eyJWljoimC4wLjAwMDAiLCJQIjoiV2luMzliLCJBTiil6k1haWwiLCJXVCi6Mn0%3D%7C1000&data=qgpqJDi2Je5lkyV%2BNyumpcMn1tk%2FTdXdWFCzsZEBKM0%3D&reserved=0>

I'm also not sure how a more moderated message would avoid an ineffective use of resources for prevention and care. Perhaps because there would be more considered dissemination of funding? I think that all gets away from your very important and unique findings, so I would encourage you to end on those.

Reply: Thank you. We have toned down the language and also removed the part on a moderated message by the media in line with the suggestions.

Page 14-15: For the tables, it would be helpful if you could highlight in bold (or whatever BMJ Open requests) any p-values the reader should take notice of as indicating a violation of an assumption of the model. That way, we can reconcile this assumption violation in the table with any description of steps taken to address it in the text. Perhaps there were no assumption violations.

Reply: Thank you for the comment. Following the suggestion from the Reviewer, we have added the corresponding sentence on model diagnostics results at the end of the "Results" section and to the footnotes on Table 1-3.

Figures 1-3: I appreciate the authors have taken steps to ease interpretation in the figures. I still think they are too busy and inconsistent with recommendations. I would therefore ask the authors to make them all consistent per:

<https://eur01.safelinks.protection.outlook.com/?url=https%3A%2F%2Fonlinelibrary.wiley.com%2Fdoi%2F10.1002%2Fjrsm.1435&data=04%7C01%7Cchristian.ruck%40ki.se%7C5815c7167262423f894108d8f9dfb2e3>

%7Cbff7eef1cf4b4f32be3da1dda043c05d%7C0%7C0%7C637534086465861227%7CUnknown%7CTWFpbGZsb

3d8eyJWljoiMC4wLjAwMDAiLCJQIjoiV2luMzliLCJBTiI6Ik1haWwiLCJXVCI6Mn0%3D%7C1000&sdata=DoO_9Bur64V9dVd4w%2F%2BZfyp5QI%2BmmmJ8io3cA15NSJrl%3D&reserved=0 (they supply the Stata code, which will work in the version of Stata you have). You won't need to plot counterfactuals, and I think the grey shading for exposures that you have is much better than a red line (which doesn't convey the period). But I think the colours as they stand are too inconsistent and not easy on the eye. I would also suggest reducing the years listed on the x-axis. Maybe every 4 or 5 years.

Reply: We are grateful for the comment and for being provided with a detailed guideline and suggestion for making the reader-friendly graphs. We have adjusted our figures according to the guidelines; although some of the features which are specifically made for the interrupted time series visualisation we did not use in order not to mislead the readers. For example, we addressed the Reviewer's suggestion on reducing the years listed on the xaxis (to every 4th year) and used the colours suggested by the guideline (e.g., colour blind friendly blue and orange). However, we did not add the pre- and post-interruption trend lines since we do not report the calculation of either level or slope changes in the text and we think that it might be misleading for the readers if the trend lines will be reported in figures. Also, it is often more difficult for the readers to visually examine and interpret the trend lines if multiple interruption points are used (as in our case there are three interruption periods). Our initial idea was to visualize the descriptive data in a simple way of reporting two time series (influenza death rates and suicide rates). Since our original figures are complicated by the use of two different Y-axes – an arithmetic scale for suicide rates and logarithmic scale for influenza death rates, we decided not to add any new elements as, for example, the time trends. Also, to ease the interpretation data on the two-y-axes figure, we did not use scatterplot, but connected the dots with a thin line. Please see the revised Figure 1 and 2 (Figure 3 was omitted since data on COVID-19 and corresponding influenza death rates and suicide rates in 2019 and 2020 were removed).

Lastly, I could not see STROBE and RECORD checklists accompanying the material, so the article would be strengthened by these populated checklists (some items in them may be n/a).

Reply: Thank you. We had submitted the STROBE checklist and now seems to be visible in the system.

Reviewer: 4

Mr. Michael Bray, Johns Hopkins Medicine, Toronto Rehabilitation Institute
Comments to the Author:

The authors are to be commended on this interesting manuscript. This article has several notable strengths, including utilization of nation-wide suicide data over multiple years (including during the COVID-19 pandemic), potentially meaningful subgroup analyses, and the timeliness of the findings.

Reply: Thank you very much.

With this in mind, I have several specific suggestions and comments, which, if integrated/addressed, could constitute a meaningful improvement to this addition to the literature:

The introduction would benefit from a more thorough review of the current literature surrounding COVID-19 and suicide mortality. Undoubtedly, the authors' review was conducted prior to the publication of some of these articles but it is essential that their findings be placed in the context of these other key findings. For instance, current data is in line with the notion that suicide mortality is not in fact increasing on the whole during the COVID19 pandemic but this is now supported by more concrete findings regarding suicide mortality than the authors cite

(see Faust et al 2021 JAMA Network Open; Appleby et al 2021 preprint; Bray et al 2020 JAMA Psychiatry; Mitchell et al 2021 Psychiatr Res; Leske et al 2021 Lancet Psychiatry). Furthermore, the authors commendable approach using subgroup analyses is supported by emerging data that some subgroups bear differing burden of suicide mortality during the COVID-19 pandemic, though the current literature has found increased suicide mortality among African Americans as opposed to gender differences (see Bray et al 2020 JAMA Psychiatry and Mitchell et al 2021 Psychiatr Res). These subgroup differences warrant comment in the introduction.

Reply: Thank you. We agree that a lot of new data now gives us a much better picture of the suicide rates during the current pandemic. This is one of the reason that we decided to remove the COVID-19 data from the manuscript. We have added some of the suggested references in the introduction. We have updated the introduction to still have a general part on COVID-19 and have included updated references, including the valuable comment on subgroup differences.

A minor comment, but the authors state in the introduction that the Spanish flu was accompanied by an increase in suicide – this is confounded by the end of World War I and warrants mention. I might recommend softer wording (for instance “the Spanish flu coincided with an increase in suicides” as opposed to “was associated with”).

Reply: As our data is Sweden only, and Sweden as well as neighboring Norway and Denmark were not part of the war, we think that the war elsewhere should probably have had minor effects on suicide.

In the introduction, more context is needed in order to quickly characterize socioeconomic fallout of other pandemics. If we are to extrapolate any of these findings to the current COVID-19 pandemic I am interested in knowing more about lockdowns, joblessness, etc. during the Spanish flu and others in Sweden.

Reply: We have now added some information on the circumstances around the Spanish Flu in Sweden in the introduction (third paragraph).

The authors report on suicides separately in men and women, while this is to be commended, your hypotheses related to the intersection of gender, pandemics and suicide are unclear. Do you expect men and women to respond differently to the stresses of the pandemic regarding suicide risk? What literature supports this?

Reply: Thank you for the opportunity to clarify. The study by Chan et al on the SARS epidemic suggested sex specific effects (Chan SMS, Chiu FKH, Lam CWL, *et al.* Elderly suicide and the 2003 SARS epidemic in Hong Kong. *Int J Geriatr Psychiatry* 2006;**21**:113–8. doi:10.1002/gps.1432). We have tried to clarify this in the introduction.

At the end of the introduction, please include any directional hypotheses that were made a priori to strengthen the impact of these findings further.

Reply: Thank you for the comment. We agree that the hypothesis on the direction of association may help the readers to interpret the findings. We have added a sentence about our initial hypothesis as “We hypothesize that such association, if any, would likely be weak”.

The dummy variable accounting for different coding of outcomes is very well thought out.

One limitation of the authors’ statistical approach is that it appears to operationalize effects of influenza deaths on suicide mortality as consistent from 1910 to 2020, without accounting for the possibility that influenza deaths during pandemics may be starkly different from those suffered outside of the context of pandemics. As such, it is difficult to draw the conclusion that suicides are not affected by pandemics based on these analyses. The authors should address this in the revised manuscript.

Reply: Thank you for the comment. We agree that it is intuitively anticipated that influenza deaths during pandemics may differ from those occurred in no-pandemic time. This was one of the challenge for us to choose the suitable statistical approach. To our understanding, prior to creating any dummy variables in relation to a possible change in exposure time series, we need to test for the presence of structural break (in our case, in a

time series of influenza deaths). In the original version of the manuscript, we relied on the results of the cumulative sum of recursive residuals and their squares (CUSUM and CUSUMQ test, respectively) which indicate whether there is a structural break due to changes in regression coefficients over time. Since in all possible models both CUSUM and CUSUMQ tests showed no structural breaks, we did not consider using any additional dummy variables that would refer to pandemic/non-pandemic periods.

After receiving the comment from the Reviewer, we additionally conducted Zivot-Andrews Unit Root test to check whether influenza death rates are sensitive to structural breaks, assuming that is such break is present a time series will abruptly change at a certain point in time. The null hypothesis of Zivot-Andrews Unit Root test is that there is a unit root in a structural break in either trend, intercept, or both. We tested all three conditions and rejected the null hypothesis in each case at 1% level. These results allowed us to conclude that influenza death rates are stationary and not sensitive to structural breaks and, thus, that influenza deaths rates do not change abruptly at any certain time point the way that it could lose structural stability and result in unreliability of the model. In light of the abovementioned arguments and the results from Zivot-Andrews tests, we consider that the creation of an additional dummy variable with, for example, pandemics years (1918, 1919, 1920, 1957, 1958, 1968, 1969) equals to 1 and the rest of the years (i.e., non-pandemic) equal to 0 may not be justified. It is also important to add that the NARDL models are sensitive to overadjustment (i.e., to the inclusion of unnecessary exogenous variables) since this may affect the model stability. Thus, we were hesitant to run an additional analysis with the extra dummy variable if the need of such variable was not clearly justified by the tests. We, however, are ready to reconsider if the Reviewer will find our

arguments to be not strong enough and insist on conducting the analysis with extra dummy variable for pandemic/non-pandemic years.

It seems to me that, during the Spanish flu, suicide rates trend upwards for the duration of the pandemic (with the exception of the first year which marks the end of World War I). I'm under the impression that this was the only of the three pandemics studied that was defined by widespread sociological changes such as lockdowns. In light of this, the degree to which we can extrapolate these results to the current pandemic is unclear, though I would appreciate the authors weighing in on this in a revised manuscript.

Reply: We have moderated the message on the degree that our results inform us about the current pandemic.

How was "short term" and "long term" operationalized? Greater clarification of this is warranted in the methods section.

Reply: In the non-linear autoregressive distributed lag (NARDL) models, the "short-term effect" refers to the **immediate** relation between independent time series (influenza death rates) and dependent time series (suicide rates), i.e., the relation can be established for each unit of time used (year, quarter, months, weeks, etc.). "Longterm effect" implies that the relationship of exposure and outcome has a lag-structure, or in other words, the effect of exposure on outcome is **distributed over a longer period of time**. In the revised article, we have briefly described the meaning of effects under the subheading "Statistical analysis" (page 4, first paragraph in the section).

When was the start of the COVID-19 pandemic in Sweden? (i.e. first case, first death, first lockdowns, etc.) It seems to me that including January in analyses as a "COVID month" may be watering down the findings by including irrelevant months if it hit in earnest during February. A description of this should be included for readers unfamiliar with the timeline of Sweden's COVID-19 course. Further, rationale for inclusion of January would be helpful.

Reply: Thank you. Since we now have removed the covid data from the manuscript this does no longer apply.

In results, the authors state that significant Wald tests may be discounted as these results "are not supported by other coefficients". While I agree that these Wald tests are not strong evidence of a relationship, explicitly stating which coefficients are not consistent might be helpful.

Reply: Thank you for the comment. We agree that clarification is needed for the sentence indicated by the Reviewer. To explain which coefficients we have in mind, we have added "... which correspond to long- and short-term associations between increases and decreases in influenza deaths and suicides among women in 1918-1956 and 1957-1978 (Tables 2 and 3)" (page 7). This means that we refer to long-term coefficients ("influenza +" and "influenza -") for the period 1918-1956 (because

Wald_{LR} was significant for women) and to both long- and short-term coefficients for the period 1957-1978 (because both Wald_{LR} and Wald_{SR} were significant for women)

There seems to be no statistical testing in the comparison of 2020 and 2019 suicide rates so it is challenging to evaluate as a reader if these are meaningful difference. Was data only available in aggregate?

Reply: Thank you. Since we now have removed the covid data from the manuscript this does no longer apply.

As stated above regarding the introduction, a more thorough review of the latest literature on COVID and suicide mortality should be considered to properly place these findings in the appropriate context. This would be helpful to demonstrate that these findings are consistent with this growing literature suggesting that suicides are not increasing in the aggregate.

In the discussion the authors mention that 1918 had the lowest suicide mortality despite highest influenza mortality. This is almost certainly confounded by the end of WWI and if this is not presented in this context it may be misleading for some readers.

Reply: Sweden was not part of the first World War so we expect quite limited effects from the war here.

The final sentence of the conclusion is quite strong, particularly on the basis of these data. A more moderate message is certainly reasonable in light of the growing literature suggesting suicide rates are not increasing in aggregate (though some sub-groups may be at elevated risk), though this conclusion is best made in the light of this growing body of literature. Dispelling moral panic at this stage is reasonable, though I would caution against advocating against allocation of resources for suicide prevention and mental health care.

Reply: We agree and have moderated the message.

I greatly appreciated the opportunity to review this interesting manuscript. Again, the authors are to be commended for taking on this potentially impactful project. I am hopeful that their consideration of the above comments will be helpful to the authors as they revise this manuscript further and I am optimistic it will constitute a meaningful contribution to the literature.

Reply: Thank you for careful reading and very helpful comments.

VERSION 2 – REVIEW

REVIEWER	Okusaga, Olaoluwa
----------	-------------------

	Baylor College of Medicine, Psychiatry and Behavioral Sciences
REVIEW RETURNED	25-May-2021

GENERAL COMMENTS	The reviewer completed the relevant checklist but made no further comment.
--

REVIEWER	Leske, Stuart Griffith University
REVIEW RETURNED	03-Jun-2021

GENERAL COMMENTS	I thank the authors for resubmitting this manuscript. I have read through your responses to the other reviewers' comments and to my comments and am satisfied with your thorough responses. It is fine to have the SDs rather than the CIs as the frequentist CIs are typically misinterpreted as Bayesian posterior intervals as the authors likely know. I could not access the STROBE but thank you for uploading it in the previous submission. The RECORD Statement is also quite applicable due to the data source (https://www.record-statement.org/), so perhaps the authors can indicate if they considered populating it or it was considered not applicable or appropriate for this study for some reason? The following comments are mostly minor and more related to language. Page 3, line 6-7: "There is concern that the COVID-19 pandemic will be associated to a surge of suicides," I find the word 'surge' a bit out of context here – perhaps the authors can say 'associated with increases in suicides' or 'associated with increased suicides'? Page 3, line 22: If wanting to keep this abstract in past tense (what you did), change 'is' to 'was'. Page 4, line 36: It's a little incorrect to say there was no support for increased suicides in Pirkis et al., as the sensitivity analysis did find increases in Puerto Rico, Japan and Austria. Perhaps 'little' or 'minimal' support might be more appropriate. Page 4, line 46: 'Increase in suicides' may read better than 'increase of suicides'. Page 5, lines 27-28: 'experienced' may read better than 'was hit'. Page 5, line 28: I'm unsure what the relevance of the 'socio-political context' is to this analysis, unless it had some bearing on the reporting of data or the way it was collected that you account for in the analysis? If it's not related to the way the data was collected (which you have accounted for), perhaps it can be omitted. I'm guessing your meaning it's related to the way the suicides being coded changed over time. Page 9, 33-38: Although this is a very safe and conservative sentence, I do wonder if the mortality rate for these pandemics differed from COVID-19, as certainly the COVID-19 mortality rate is lower than SARS. Since the suicide rate is still viewed alongside
--

	influenza deaths, differing mortality rates may warrant some mention here. Page 10, lines 11-12: I again wonder if you can use some language in the conclusion indicating the relative mortality rate of these influenza pandemics (vs SARS, COVID) to make the concluding statement more applicable to the pandemics you studied? One would hope that if there is no appreciable increase in pandemics with higher mortality rates, there would not be so in one with lower mortality rates. Page 10, line 47: There is a typo – mortality. I would say on the whole though this statement is not applicable for other reasons as you have earlier indicated there are no participants and patient organizations and you cannot disseminate data to deceased individuals. 'Groups' just needs an apostrophe if you retain this statement. One remaining inconsistency throughout the manuscript is that you refer to many factors in the introduction but then don't go on to study these. I do acknowledge that I suggested citing some of this material. For instance, there is social distancing and disconnectedness, an economic downturn, the decreased access to mental health services, increased access to lethal means, unemployment rates, recession, and increased impacts on elderly people (seems to be 65 years and older). I acknowledge looking at all these is beyond the scope of this manuscript, which is answering the more basic and fundamental research question and stratifying by sex. So I think it might be best if you just add to the limitations section saying you did not look at these things but now that you have progressed beyond answering this fundamental RQ, there is the opportunity to look at some of these RQs. Elderly adults is perhaps the most pressing topic since there is previous evidence they are adversely impacted in terms of suicides in pandemics/SARS and age group and sex are typically basic confounders. If you consider this all not a limitation, perhaps at least add it to future research for your group. Also, there is some American spelling throughout (z instead of s). Lastly, I thank the authors for providing their dataset and Stata code in the interests of open science and so that people wishing to analyse their own data using a similar approach can do so.
--	--

REVIEWER	Bray, Michael Johns Hopkins Medicine, Institute of Medical Science
REVIEW RETURNED	24-May-2021
GENERAL COMMENTS	The authors are to be commended on their excellent and thoughtful revisions. I have no further comments.

VERSION 2 – AUTHOR RESPONSE

Reviewer: 2

Dr. Olaoluwa Okusaga, Baylor College of Medicine, Michael E DeBakey VA Medical Center

Comments to the Author:

None

Reviewer: 3

Dr. Stuart Leske, Griffith University

Comments to the Author:

I thank the authors for resubmitting this manuscript. I have read through your responses to the other reviewers' comments and to my comments and am satisfied with your thorough responses.

Response: Thank you!

It is fine to have the SDs rather than the CIs as the frequentist CIs are typically misinterpreted as Bayesian posterior intervals as the authors likely know.

I could not access the STROBE but thank you for uploading it in the previous submission. The RECORD Statement is also quite applicable due to the data source (<https://eur01.safelinks.protection.outlook.com/?url=https%3A%2F%2Fwww.record-statement.org%2F&data=04%7C01%7Cchristian.ruck%40ki.se%7C8850f07bae6e4a71446408d93003f180%7Cbff7eef1cf4b4f32be3da1dda043c05d%7C0%7C0%7C637593615764626671%7CUnknown%7CTWFpbGZsb3d8eyJWljiMC4wLjAwMDAiLCJQIjoiV2luMzliLCJBTiI6Ikl1haWwiLCJXVCi6Mn0%3D%7C1000&sdata=wyopFlx8NKNgVumB5Ko41qcthOLns0y7LnFvruAxseM%3D&reserved=0>), so perhaps the authors can indicate if they considered populating it or it was considered not applicable or appropriate for this study for some reason?

Response: Thank you. We await the orders of the editorial staff on whether the uploaded STROBE will suffice.

The following comments are mostly minor and more related to language.

Page 3, line 6-7: "There is concern that the COVID-19 pandemic will be associated to a surge of suicides," I find the word 'surge' a bit out of context here – perhaps the authors can say 'associated with increases in suicides' or 'associated with increased suicides'?

Response: Thank you for taking the time to improve this manuscript. Really appreciated. We have changed according to the suggestions.

Page 3, line 22: If wanting to keep this abstract in past tense (what you did), change 'is' to 'was'.

Response: Thank you, we have changed to "was".

Page 4, line 36: It's a little incorrect to say there was no support for increased suicides in Pirkis et al., as the sensitivity analysis did find increases in Puerto Rico, Japan and Austria. Perhaps 'little' or 'minimal' support might be more appropriate.

Response: Changed to "minimal support".

Page 4, line 46: 'Increase in suicides' may read better than 'increase of suicides'.

Response: Thank you very much, changed as suggested.

Page 5, lines 27-28: 'experienced' may read better than 'was hit'.

Response: Thank you very much, changed as suggested.

Page 5, line 28: I'm unsure what the relevance of the 'socio-political context' is to this analysis, unless it had some bearing on the reporting of data or the way it was collected that you account for in the analysis? If it's not related to the way the data was collected (which you have accounted for), perhaps it can be omitted. I'm guessing your meaning it's related to the way the suicides being coded changed over time.

Response: the reason for the wording is to point out the fact that the study included data from different time periods, something we think strengthens the conclusions.

Page 9, 33-38: Although this is a very safe and conservative sentence, I do wonder if the mortality rate for these pandemics differed from COVID-19, as certainly the COVID-19 mortality rate is lower than SARS. Since the suicide rate is still viewed alongside influenza deaths, differing mortality rates may warrant some mention here.

Response: Thank you. We have added mortality rates to the sentence.

Page 10, lines 11-12: I again wonder if you can use some language in the conclusion indicating the relative mortality rate of these influenza pandemics (vs SARS, COVID) to make the concluding statement more applicable to the pandemics you studied? One would hope that if there is no appreciable increase in pandemics with higher mortality rates, there would not be so in one with lower mortality rates.

Response: Thank you. We believe that the sentence works in the current for but are open to change our mind.

Page 10, line 47: There is a typo – mortality. I would say on the whole though this statement is not applicable for other reasons as you have earlier indicated there are no participants and patient organizations and you cannot disseminate data to deceased individuals. 'Groups' just needs an apostrophe if you retain this statement.

Response: thank you spotting these typos. Our understanding is that this section is required.

One remaining inconsistency throughout the manuscript is that you refer to many factors in the introduction but then don't go on to study these. I do acknowledge that I suggested citing some of this material. For instance, there is social distancing and disconnectedness, an economic downturn, the decreased access to mental health services, increased access to lethal means, unemployment rates, recession, and increased impacts on elderly people (seems to be 65 years and older). I acknowledge looking at all these is beyond the scope of this manuscript, which is answering the more basic and fundamental research question and stratifying by sex. So I think it might be best if you just add to the limitations section saying you did not look at these things but now that you have progressed beyond answering this fundamental RQ, there is the opportunity to look at some of these RQs. Elderly adults is perhaps the most pressing topic since there is previous evidence they are adversely impacted in terms of suicides in pandemics/SARS and age group and sex are typically basic confounders. If you consider this all not a limitation, perhaps at least add it to future research for your group.

Response: We have tried to put the data in a current context and that means that we have covered some issues regarding the ongoing pandemic that are not part of our analyses but in our opinion still provide useful context.

Also, there is some American spelling throughout (z instead of s).

Response: Thanks for finding those!

Lastly, I thank the authors for providing their dataset and Stata code in the interests of open science and so that people wishing to analyse their own data using a similar approach can do so.

Response: Thank you!

Reviewer: 4

Mr. Michael Bray, Johns Hopkins Medicine, Toronto Rehabilitation Institute

Comments to the Author:

The authors are to be commended on their excellent and thoughtful revisions. I have no further comments.

Response: Thank you!